# In vivo intraoral waterflow quantification reveals hidden mechanisms of suction feeding in fish

Pauline Provini[1,2]*, Alexandre Brunet[1], Andréa Filippo[1], Sam Van Wassenbergh[1,3]

[1]Département Adaptations du Vivant, Paris, France; [2]Université de Paris, INSERM U1284, Center for Research and Interdisciplinarity (CRI), Paris, France; [3]Department of Biology, University of Antwerp, Universiteitsplein 1, Antwerp, Belgium

**Abstract** Virtually all fishes rely on flows of water to transport food to the back of their pharynx. While external flows that draw food into the mouth are well described, how intraoral waterflows manage to deposit food at the esophagus entrance remains unknown. In theory, the posteriorly moving water must, at some point, curve laterally and/or ventrally to exit through the gill slits. Such flows would eventually carry food away from the esophagus instead of toward it. This apparent paradox calls for a filtration mechanism to deviate food from the suction-feeding streamlines. To study this gap in our fundamental understanding of how fishes feed, we developed and applied a new technique to quantify three-dimensional (3D) patterns of intraoral waterflows in vivo. We combined stereoscopic high-speed X-ray videos to quantify skeletal motion (XROMM) with 3D X-ray particle tracking (XPT) of neutrally buoyant spheres of 1.4 mm in diameter. We show, for carp (*Cyprinus carpio*) and tilapia (*Oreochromis niloticus*), that water tracers displayed higher curvatures than food tracers, indicating an inertia-driven filtration. In addition, tilapia also exhibited a 'central jet' flow pattern, which aids in quickly carrying food to the pharyngeal jaw region. When the food was trapped at the branchial basket, it was resuspended and carried more centrally by periodical bidirectional waterflows, synchronized with head-bone motions. By providing a complete picture of the suction-feeding process and revealing fundamental differences in food transport mechanisms among species, this novel technique opens a new area of investigation to fully understand how most aquatic vertebrates feed.

*For correspondence: pauline.provini@cri-paris.org

**Competing interest:** The authors declare that no competing interests exist.

## Editor's evaluation

How do fish suck food underwater? Using new artificial food particles that are radio-opaque and naturally buoyant, Provini et al. imaged the roller-coaster ride that food particles make being sucked in from outside to inside the fish using 3D stereo high-speed fluoroscopy. The recordings show fishes to have an intriguing ability to generate flows that center the food particles as they enter the muscular tube that carries them from the mouth to the stomach. Remarkably, the flow patterns in the mouth that accomplish this seem to differ between the two species of fish studied, although samples sizes are very small at present and results should be interpreted cautiously. These new insights will be of interest to biologists working on suction-feeding mechanisms ranging from millimeter-sized carnivorous water plants, tadpoles, and fish larvae, to large fish and marine mammals, and even gigantic whales. Bioinspired engineers designing rapid underwater suction apparatuses may benefit from harnessing the new insights to elegantly center items of interest.

**Figure 1.** Hypothetical waterflow patterns during suction feeding in fish. (**a**) Schematic illustration of extraoral flows (blue path line arrows) and hypothetical intraoral flows (red path line arrows) with important anatomical structures involved in food interception (the right-side buccopharyngeal walls and branchial basket are removed and only left-side path lines are shown) in a bony fish. (**b–d**) Three hypotheses for intraoral flow patterns, in a dorsoventral view. Note that from (**b**) to (**d**), the tendency to move food away from the desired food target, the esophagus, decreases.

## Introduction

How organisms acquire food is largely determined by the medium in which they feed (*Herrel et al., 2012*). Aquatic organisms commonly exploit the dense and viscous properties of water to carry food from a distance, toward and through their mouth, by generating flows of water. These flows typically result from suction created by a powerful expansion of the feeding apparatus, for example, in fishes that suddenly increase the volume of their buccopharyngeal cavity (*Alexander, 1969*). This action is referred to as "suction feeding".

A wide range of aquatic vertebrates across a large size range employ suction feeding: from larval fishes (*Drost et al., 1988*) and frog tadpoles (*Deban and Olson, 2002*) to whales (*Werth, 2004*), and some carnivorous plant with tiny suction traps (*Müller et al., 2020*). Because of this impressive diversity in organisms making use of suction feeding, it plays an important role in all major aquatic habitats and feeding niches throughout the water column (*Wainwright et al., 2015*).

In virtually all fishes, waterflows generated by suction are essential for transporting food inside the buccal cavity (*Labarbera, 1984*; *Potvin and Werth, 2017*; *Wainwright et al., 2015*). During suction feeding, the three-dimensional (3D) waterflow patterns toward the fish's mouth have been particularly well studied (*Day et al., 2007*; *Nauwelaerts et al., 2008*; *Skorczewski et al., 2012*; *van Leeuwen, 1984*). Although waterflow patterns external to the mouth depend on the approaching speed of the fish (*Kane and Higham, 2014*; *Muller and Osse, 1984*), instantaneous streamline patterns are fairly conserved between species and behaviors (*Jacobs and Holzman, 2018*): as the 3D streamlines converge toward the gape (*Figure 1a*, blue arrows), food items that are initially somewhere close to the mouth will curve into the buccal cavity by these extraoral waterflows.

In contrast to this relatively straightforward and well-resolved mechanism of food capture toward the mouth, food transport mechanisms inside the buccopharyngeal cavity are not as simple (*Callan and Sanderson, 2003*; *Divi et al., 2018*; *Sanderson et al., 1991*; *van Meer et al., 2019*) and are largely unknown (*Day et al., 2015*). After passing through the mouth opening (*Figure 1a*, blue arrows), water streamlines are generally assumed to diverge along with the widening buccopharynx (*Figure 1a and b*, red arrows; *Muller et al., 1985*; *Provini and Van Wassenbergh, 2018*). This implies that, inside the buccal cavity, waterflow tends to be directed toward the gill slits, located at the posteroventral and posterolateral head margins (*Figure 1b*). Paradoxically, as these flows take a short route to the gill slits, they seem far from optimal to carry food to the entrance of the esophagus, along the midsagittal plane of the buccal cavity.

To date, there is no evidence that the esophagus expands in concert with buccopharyngeal suction generation to directly draw in a significant amount of water together with the food. Therefore, other structures need to be involved in separating food from the waterflows. Ideally, in bony fish, the food that is sucked into the mouth could be intercepted by the pharyngeal jaws located just anterior to the esophageal sphincter (*Figure 1*). The mobile pharyngeal jaws, while maintaining physical contact with the food, can move it to the esophagus (*Liem, 1978*; *Sibbing et al., 1986*). However, given the current hypothesis of diverging streamlines in the expanding buccopharyngeal cavity (*Figure 1a and b*), food is more likely to end up being sieved by the other branchial arches and their gill rakers, after which a second food transport cycle is needed, encountering the same problem regarding how the food is able to reach the esophagus. Consequently, intraoral hydrodynamics are critical in determining the initial food deposit site after capture and, in turn, the associated anatomical adaptations and behaviors for food transport and handling inside the head.

Alternatively, two other intraoral flow patterns are theoretically possible, which may facilitate interception by the pharyngeal jaws (*Figure 1c and d*). Computational models of suction-feeding hydrodynamics (*Thompson et al., 2018*; *Van Wassenbergh, 2015*) show that flow can separate from the expanding buccopharynx to create a centralized jet of suction flow, along with anterior flow at the borders of the buccal cavity forming vortices (*Figure 1c*). This pattern, to which we will refer as a "central jet", could carry food closer to the pharyngeal jaws (*Figure 1c*) compared to the classical hypothesis of "diverging flow" (*Figure 1b*). Thirdly, as hypothesized for fishes relying on filtration of smaller food particles from the water (*Brooks et al., 2018*; *Sanderson et al., 2001*), gill arch and raker structures may cause crossflows, in which the main direction of flow medial to the branchial arches is directed toward the esophagus and only the permeate of the branchial filter basket flows out through the gill slits (*Figure 1d*).

Despite this critical role during feeding in nearly all fishes, the spatiotemporal pattern of intraoral waterflow has still not been quantified in vivo due to the difficulty in gaining optical access to the buccopharyngeal cavity. This implies a fundamental gap in our knowledge regarding how more than half of all vertebrates feed. Here, we develop a new technique based on biplanar high-speed X-ray videography to simultaneously quantify the 3D path lines of both food and intraoral water, as well as skeletal kinematics. We apply this technique to two distantly related omnivorous suction feeders, carp (*Cyprinus carpio*) and tilapia (*Oreochromis niloticus*). These species were chosen from the large diversity of suction feeders because their adult head size is small enough to fit the imaging volume of the biplanar X-ray setup, yet large enough to avoid hindrance from an intake of tracer particles for waterflow inside the buccal cavity using X-rays. We use this data to describe the intraoral waterflow and determine the intraoral mechanisms of aquatic feeding that have remained hidden thus far.

## Results

### Intraoral kinematics of water tracers and food

#### The periodic bidirectional motion of water tracers

During each feeding sequence (carp: two individuals, seven trials; tilapia: two individuals, six trials), the water tracers moved alternatingly backward and forward, interspersed with phases of stasis (*Figure 2*, *Figure 2—video 1*, *Figure 2—video 2*). At the beginning of the sequence, for both species, we observed a relatively slow posterior motion of the water tracers toward the posterior part of the buccal cavity (*Figure 2*). As the velocity of the water tracers is calculated in reference to the entrance

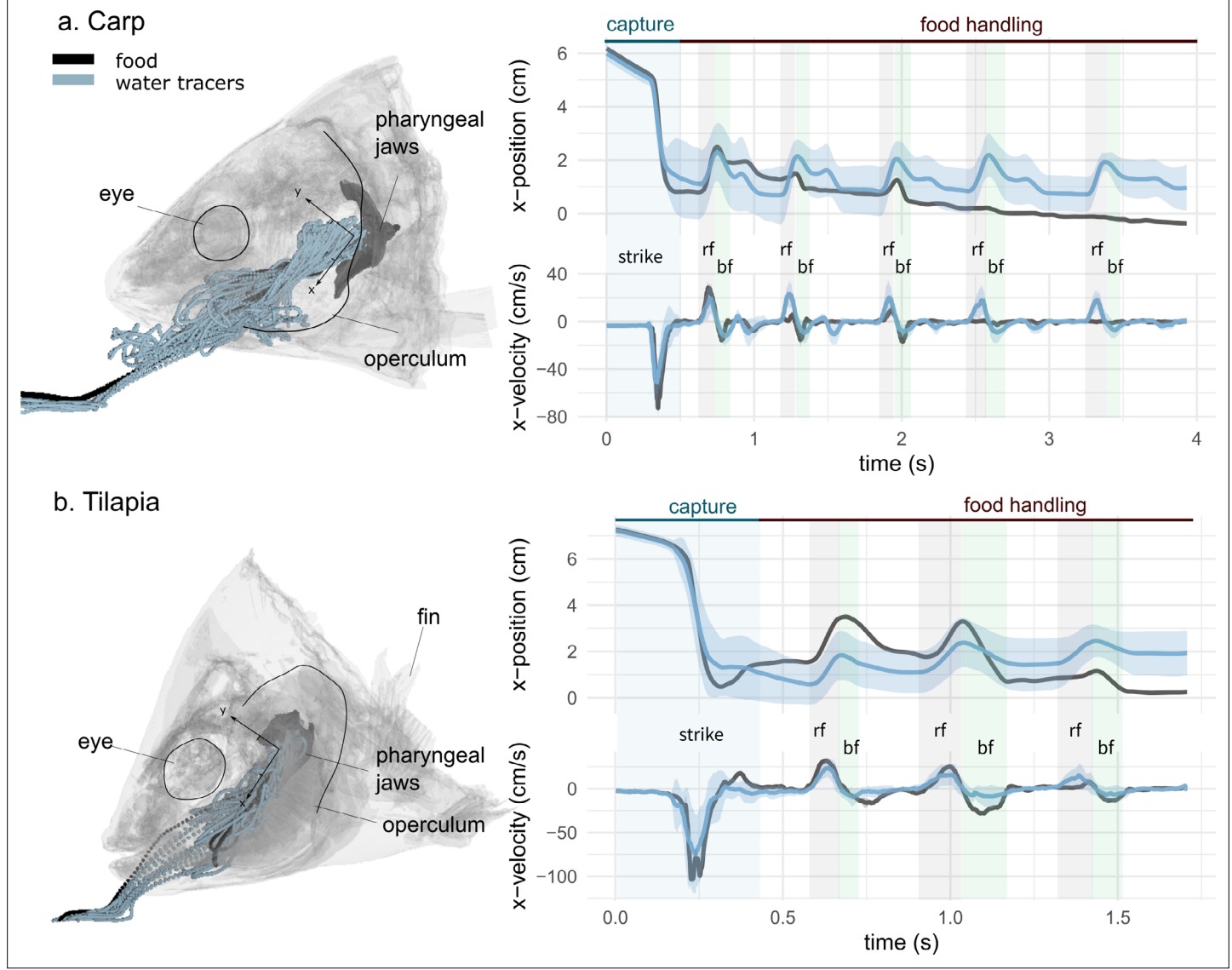

**Figure 2.** Representative sequences of suction feeding for carp (**a**) and tilapia (**b**). Water and food tracer 3D trajectories outside and inside the buccal cavity are represented by a semi-transparent model of a carp (**a**, left) and a tilapia (**b**, left) head. The anteroposterior velocity of the water and food tracers of a carp (**a**, right) and a tilapia (**b**, right) over a representative suction-feeding sequence are shown, with the mean trajectory of the water tracers (blue line) surrounded by the standard deviation (lighter blue), as well as the trajectory of the food tracer (black). On the graphs, we show the division into two main phases: capture and intraoral food handling. During capture, the intake corresponds to a high negative magnitude for the anteroposterior velocity, followed by stasis, during which the anteroposterior velocity is close to zero. During intraoral food handling, we see an alternance of stasis consisting of a reverse flow (rf) with a positive anteroposterior velocity and of a backflow (bf) with a negative anteroposterior velocity of lower amplitudes compared to the first strike. (carp: two individuals, seven trials; tilapia: two individuals, six trials). The presented trials correspond to the X-ray videos (*Figure 2—video 1*, *Figure 2—video 2*).

The online version of this article includes the following video, source data, source code, and figure supplement(s) for figure 2:

**Source data 1.** Normalized displacement components along the three cranium-bound orthogonal axes shown in *Figure 2* (mean ± SD in %) for the food and water tracers in carp and tilapia.

**Source code 1.** R code generating the graphs of *Figure 2*.

**Source code 2.** R code identifying the phases during a suction-feeding sequence used in *Figure 2*.

**Figure supplement 1.** Histograms of the total distance covered by the food (in black) and the water tracers (in blue) during an entire sequence of suction feeding for each carp (**a**) and tilapia (**b**) trial.

**Figure supplement 1—source code 1.** R code generating *Figure 2—figure supplement 1*.

**Figure 2—video 1.** Video of a suction-feeding sequence of a carp, showing the raw video, the water and food tracer trajectories, and the rigid-body

*Figure 2 continued*
reconstruction.
https://elifesciences.org/articles/73621/figures#fig2video1
**Figure 2—video 2.** Video of a suction-feeding sequence of a tilapia, showing the raw video, the water and food tracer trajectories, and the rigid-body reconstruction.
https://elifesciences.org/articles/73621/figures#fig2video2

of the esophagus, this low velocity corresponds to the approaching speed of the fish. It corresponds to 3.9 ± 2.5 cm/s in carp and 3.3 ± 1.3 cm/s in tilapia (*Figure 2—source data 1*).

Then, we observed a rapid increase in the posterior velocity of the water tracers, reaching up to 92.1 ± 59 cm/s for the carp and 97.8 ± 23 cm/s for the tilapia. We named this rapid increase "intake" (*Figure 2*), corresponding to what is commonly known as the food-capture or food-intake phase. For the carp and tilapia, this phase lasted 0.26 ± 0.04 s and 0.27 ± 0.08 s, respectively.

After the intake, a phase of apparent stasis occurred, with a steady, low-velocity, anteroposterior water tracer trajectory, leading to a nearly complete stop of the food.

Then, the water tracers started to move again, corresponding to the beginning of the intraoral food-handling phase. This phase was characterized by lower velocities compared to the capture phase (the maximum during food capture was 32.9 ± 11 cm/s in carp and 24.7 ± 12 cm/s in tilapia) and the occurrence of bidirectional motion: reverse flows (rfs), defined by negative anteroposterior velocities of the water tracers (highlighted in gray in *Figure 2*), and backflows (bfs), with a similar but slower pattern than the intake (highlighted in green in *Figure 2*).

The succession of the intake and the alternating rf and bf phases were consistent across trials and could be automatically detected in the studied trials (carp: two individuals, seven trials; tilapia: two individuals, six trials) (*Figure 2—source code 2*). We found that the periodicity of back-and-forth motions (rf and bf) was consistent throughout the sequence. For the carp, the transition between the rf and bf phases happened every 0.58 ± 0.12 s; for the tilapia, it happened every 0.40 ± 0.05 s.

## Differences between the water tracer and food trajectories

Among the 73 trials recorded, no water tracers passed the esophagus; however, the food was successfully ingested in most cases (5 and 6 failures or spit outs of 34 and 36 trials in carp and tilapia, respectively). We were able to fully analyze (hydrodynamics and bone kinematics analyses) seven trials for carp and six trials for tilapia. Among these trials, the total distance traveled by the food tracers during the pooled intake, the rf, and the bf phases was significantly lower than the distance covered by the water tracers (*Figure 2—figure supplement 1*, *Figure 2—figure supplement 1—source code 1*), demonstrating a different path between the food and water tracers.

The 3D radio-opaque particle-tracking method allowed us to export the 3D trajectory of all the water tracers during a suction-feeding sequence. This provided us with a visualization of the water tracer paths for both the lateral and dorsoventral views during the intake (*Figure 3a*), the rf phase (*Figure 3b*), and the bf phase (*Figure 3c*). To avoid overloading the figure, we chose to show a representative trial and the combination of three trials for the individuals for which, overall, particle tracking distributed best over the buccopharynx (carp 02 and tilapia 01). The data for the other individuals of carp and tilapia is available in (*Figure 3—figure supplement 1*).

During the intake, the food item usually followed a trajectory that was more dorsal in comparison to the water tracer trajectory in both species (*Figure 3a*, *Figure 3—figure supplement 1*). In addition, the food trajectory remained close to the midsagittal plane, whereas the water tracers tended to spread throughout the entire buccal cavity (*Figure 3a*, *Figure 3—figure supplement 1*). Among the dozens of water tracers captured by the fish during each trial, only 3–5 per trial also followed a midsagittal trajectory to reach a position close to the entrance of the esophagus (*Figure 3a*, *Figure 3—figure supplement 1*). The others spread to the lateral parts of the buccal cavity and stopped at the gill rakers (*Figure 3a*). In tilapia, the water tracer deviation started more anteriorly (*Figure 3a*, *Figure 3—figure supplement 1*), but in both species (carp: two individuals, seven trials; tilapia: two individuals, six trials), the path curvature of the water tracers was significantly higher than that of the food tracers (*Figure 4*), demonstrating a more important lateral deviation of the water tracers compared to the food tracers.

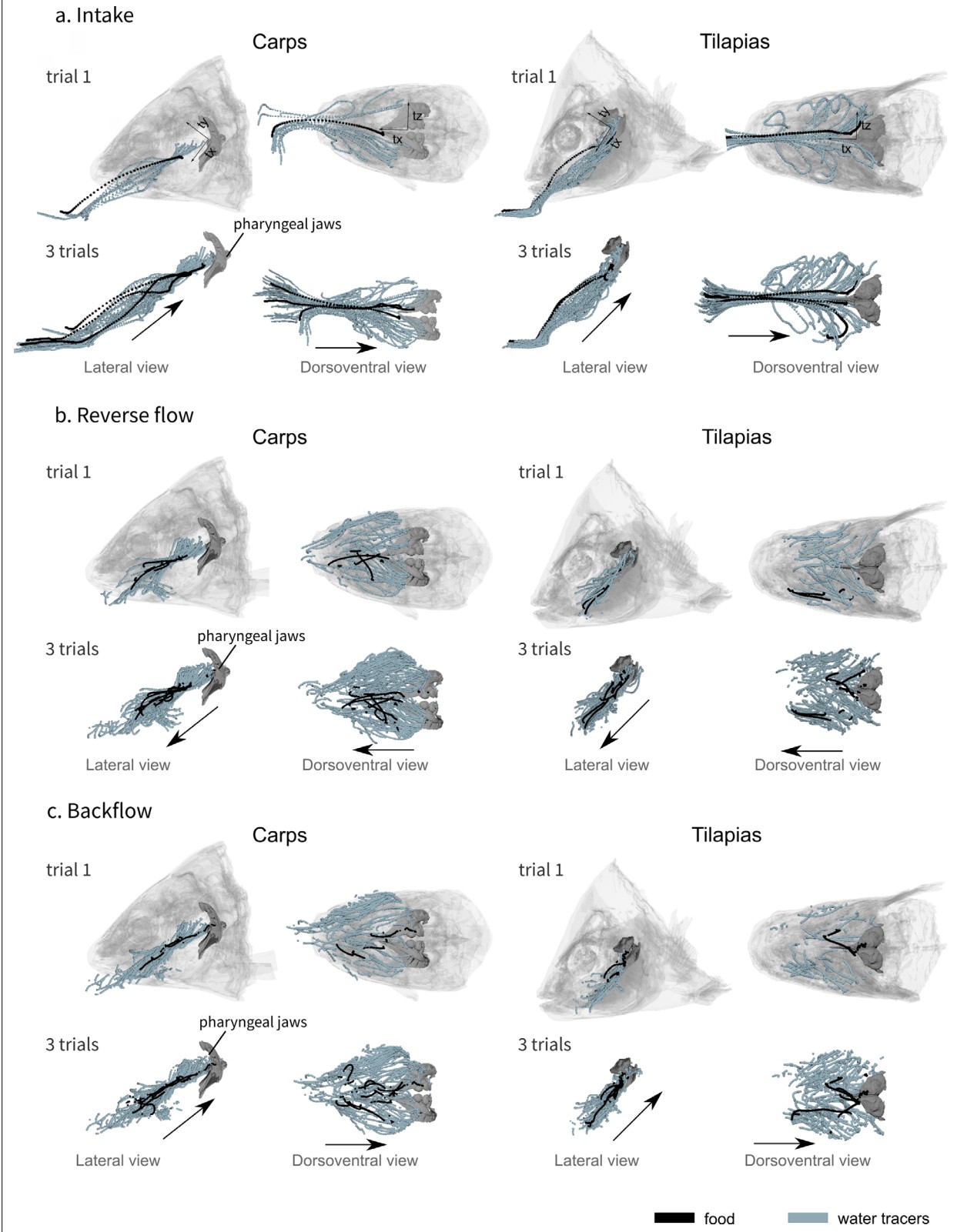

**Figure 3.** Visualization of the water tracer paths for both the lateral and dorsoventral views during the intake (**a**), the reverse flow (**b**), and the backflow (**c**). A representative trial is presented on the first row of each phase and a combination of three trials for one individual of carp (carp 02) and three trials for one individual of tilapia (tilapia 01) on the second row of each phase, showing the consistency of the patterns. The axes are represented on the first

*Figure 3 continued on next page*

*Figure 3 continued*

row of schematics, the orientation remains the same for all the other schematics. The black arrows represent the global direction of tracer movement. The data for the other individuals of carp and tilapia is available in *Figure 3—figure supplement 1*.

The online version of this article includes the following figure supplement(s) for figure 3:

**Figure supplement 1.** Visualization of the water tracer paths for both the lateral and dorsoventral views during the intake (**a**), the reverse flow (**b**), and the backflow (**c**).

---

During the other phases, the water tracers spread throughout the entire buccal cavity, but their trajectory remained predominantly anteroposterior (*Figure 2—figure supplement 1*, *Figure 3b and c*). After the first part of the intake, there is not a perfect match between the food and the tracers' trajectory, but the food moves in the same direction as the water tracers (*Figure 2*, *Figure 2—video 1*, *Figure 2—video 2*, *Figure 3*, *Figure 3—figure supplement 1*). This suggests that the food was passively moved by the waterflows, which eventually managed to reposition the food near the esophagus.

## Intake flow patterns

Especially for tilapia, at the end of the intake, many flow tracers moved anteriorly after starting to curve in the lateral direction (*Figure 4b*). Such anterior flow, close to the borders of the buccal cavity, is characteristic of the central jet pattern (*Figure 1c*). In contrast to tilapia, in carp such flows occur only close to the mouth entrance (*Figure 4a*), while flow in the posterior half of the buccal cavity resembles a diverging flow pattern (*Figure 1b*). Posteriomedial flows in the branchial arch region, which would be indicative of crossflow (*Figure 1d*), could not be discerned. In carp, only a few of the many flow tracer paths were directed medially (see green tracers in *Figure 4a*). In tilapia, none of the paths showed a clear medial component (*Figure 4b*).

We found no correlation between the moment a tracer was ingested and its final location in the buccal cavity. For both species, the Kendall test of correlation between the location at the moment of ingestion (tz at the beginning of intake) and the final location of the particle (tz at the end of intake) was not significant (carp: p-value=0.27; tilapia: p-value=0.21). However, visually, it seems that early-sucked particles tend to stay in the medial plane of the buccal cavity, participating to the central jet, whereas the late ones tend to end up on the lateral parts of the buccal cavity (*Figure 4—figure supplement 1*, *Figure 4—figure supplement 1—source code 1*).

Generally, the particles tend to stay on the same side as their initial location, but there can be a change of side before the particles arrive close to the esophagus (*Figure 4—figure supplement 2*, *Figure 4—figure supplement 2—source code 1*). A Kendall test of correlation revealed a significant correlation between the lateromedial location at the beginning of the intake phase (tz) and the latero-medial location at the end of the intake (tz) in carp but not in tilapia (carp: p-value=0.000287; tilapia: p-value=0.07952).

## Kinematics of cranial bones

The 3D kinematic analysis of the head bones revealed a stereotypical sequence of motions of the upper and lower jaws (gape), hyoid depression, and opercula abduction (carp: two individuals, seven trials; tilapia: one individual, three trials). During the intake (highlighted in blue in *Figure 5*), a peak gape was followed by a peak in hyoid depression, followed by a peak in opercula abduction. This sequence of peaks occurred in each trial for both species (*Figure 5—source code 1*). The abduction of the opercula is prominent at the time of lateral motion of the water tracers after they enter the buccal cavity. The kinematics during the bf phases corresponded to the same pattern of motions, but with lower magnitudes for the peak gape and hyoid depression than during the intake. During the rf phase (highlighted in gray in *Figure 5*), the sequence of peaks was inverted with respect to the intake: a peak in the opercula opening was followed by a peak in hyoid depression, followed by a peak gape. This reverse sequence of peaks, when compared with the intake, occurred in each trial for both species (*Figure 5—source code 1*).

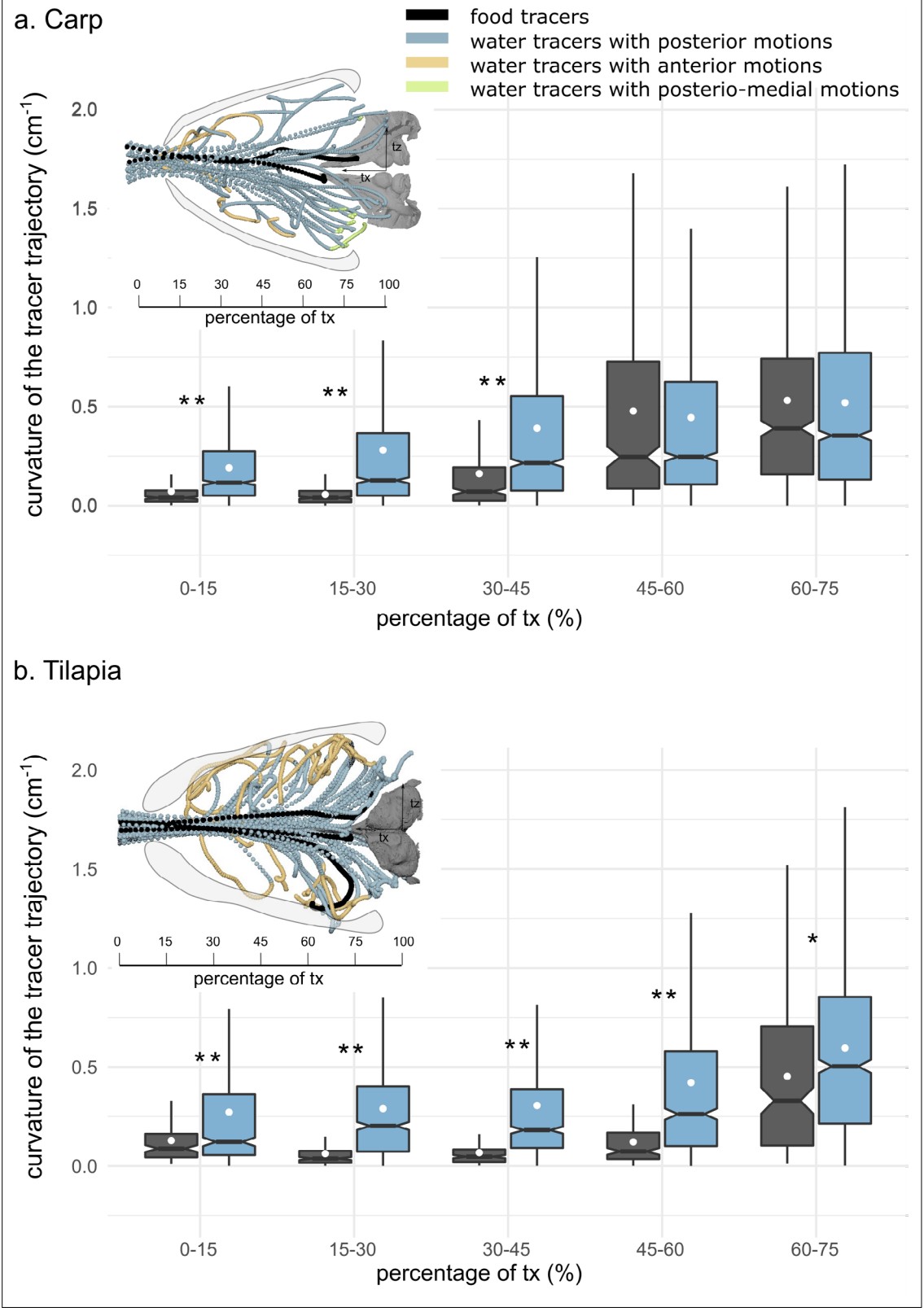

**Figure 4.** Boxplots of the mediolateral curvature calculated based on intervals of the anteroposterior trajectory during the intake in carp (**a**) and tilapia (**b**). The lower and upper hinges correspond to the first and third quartiles (the 25th and 75th percentiles), and the white dot gives a 95% confidence interval for comparing medians. The asterisks represent the significance of the statistical tests comparing the water and food tracers' curvatures (*p-value≤0.05, **p-value≤0.01) (carp: two individuals, seven trials; tilapia: two individuals, six trials). The schematics in the upper part of each subfigure

*Figure 4 continued on next page*

*Figure 4 continued*

correspond to a dorsoventral view of three trials of one individual of carp (**a**) and three trials of one individual of tilapia (**b**) showing the water (in blue, yellow, and green) and food tracer (in black) trajectories. The blue, yellow, and green trajectories highlight the posterior, anterior, and posteriomedial motions of the water tracers, respectively.

The online version of this article includes the following source code and figure supplement(s) for figure 4:

**Source code 1.** R code generating the graphs of *Figure 4*.

**Figure supplement 1.** Visualization of the water tracer trajectories on a dorsoventral view (tx, tz) during the intake for the seven trials of carp 01 and carp 02 (**a**) and the six trials of tilapia 01 and tilapia 02 (**b**).

**Figure supplement 1—source code 1.** R code generating the graphs of *Figure 4—figure supplement 1*.

**Figure supplement 2.** Visualization of the water tracer trajectories on a dorsoventral view (tx, tz) during the intake for the four trials of carp 01 (**a**), the three trials of carp 02 (**b**), the three trials of tilapia 01 (**c**), and the three trials of tilapia 02 (**d**).

**Figure supplement 2—source code 1.** R code generating the graphs of *Figure 4—figure supplement 2*.

## Discussion

This first application of neutrally buoyant X-ray particle tracking in biological research allowed us to quantify 3D waterflow patterns inside the mouth of a live fish with satisfactory spatial and temporal resolution (time resolution: 750 fps; spatial resolution of the biplanar X-ray device: ± 0.1 mm; *Sanctorum et al., 2019*). Despite the inevitably low number of tracked particles (<20) compared to common, light-optical techniques using dense suspensions of considerably smaller seeding particles (e.g., particle-tracking velocimetry), we managed to resolve the flow speeds and directions in the vicinity of the food. Compared to previously employed invasive techniques to quantify intraoral flows, such as endoscopy (*Callan and Sanderson, 2003*; *Smith and Sanderson, 2008*) or pressure recordings (e.g., *Van Leeuwen and Muller, 1982*), our noninvasive approach yields a more complete view of the spatial aspects of intraoral suction-feeding dynamics, allowing us to demonstrate the existence of different hydrodynamic mechanisms for food transport.

The new particle-tracking technique we described here (Figure 7a–d) can be easily combined with 3D motion reconstruction protocols for skeletal elements (i.e., X-ray reconstructions of the moving morphology [XROMM]; *Brainerd et al., 2010*) to link waterflow patterns to skeletal kinematics (*Figure 5*). The relatively straightforward, inexpensive, yet time-consuming protocol to construct the particles, as proposed in this study (Figure 7a–d), may be further optimized to reduce the variability in density (Figure 7d). That way, particles could be kept in suspension in the water column for longer periods and allow an accurate tracking for a wider range of waterflows, including those involving less strong pressure gradients than during suction feeding. Mold-based manufacturing protocols for metal-cored expanded polystyrene (EPS) spheres, currently available for relatively large particles with a diameter of 8 mm (*Drake et al., 2011*), could perhaps be miniaturized. In any case, the method of X-ray particle tracking has strong potential to become an indispensable tool for studying form–function relationships and the biomechanics of various functions involving water motions to which we lack optical access.

The main question of our study was how food ends up at the esophagus entrance when the water that carries the food is traveling elsewhere – namely, to the outflow at the gill slits. If water was a pure "carrier" of the food, it would follow our water tracer's trajectory from the entrance of the buccal cavity to end up in physical contact with the gill rakers (*Figure 1*). However, both carp and tilapia appeared to be well capable of positioning the food close to the midsagittal streamlines, thereby avoiding the food from becoming laterally deviated and ending up adhering to the branchial basket. Despite the fact that only a small proportion of the water path lines passed close to the pharyngeal jaws (*Figure 3*), direct interception of the food item by the pharyngeal jaw was common, which avoided the need for subsequent intraoral food-handling actions.

We hypothesize that fish make use of two mechanisms to cause the food to follow a midsagittal trajectory toward the esophagus: (1) developing a "central jet" flow pattern inside the buccal cavity (*Figure 1c*) and (2) exploiting the food's inertia and finite size. Our study found support for the first mechanism by showing a centrally concentrated flow with reverse flow developing near the borders of the buccal cavity (*Figures 4 and 6*, *Figure 6—video 1*). This central jet (*Figure 1c*) was previously reported in computational modeling studies (*Thompson et al., 2018*; *Van Wassenbergh, 2015*; *Figure 6*), but was not given any attention thus far. It was thought to be an artifact of not having

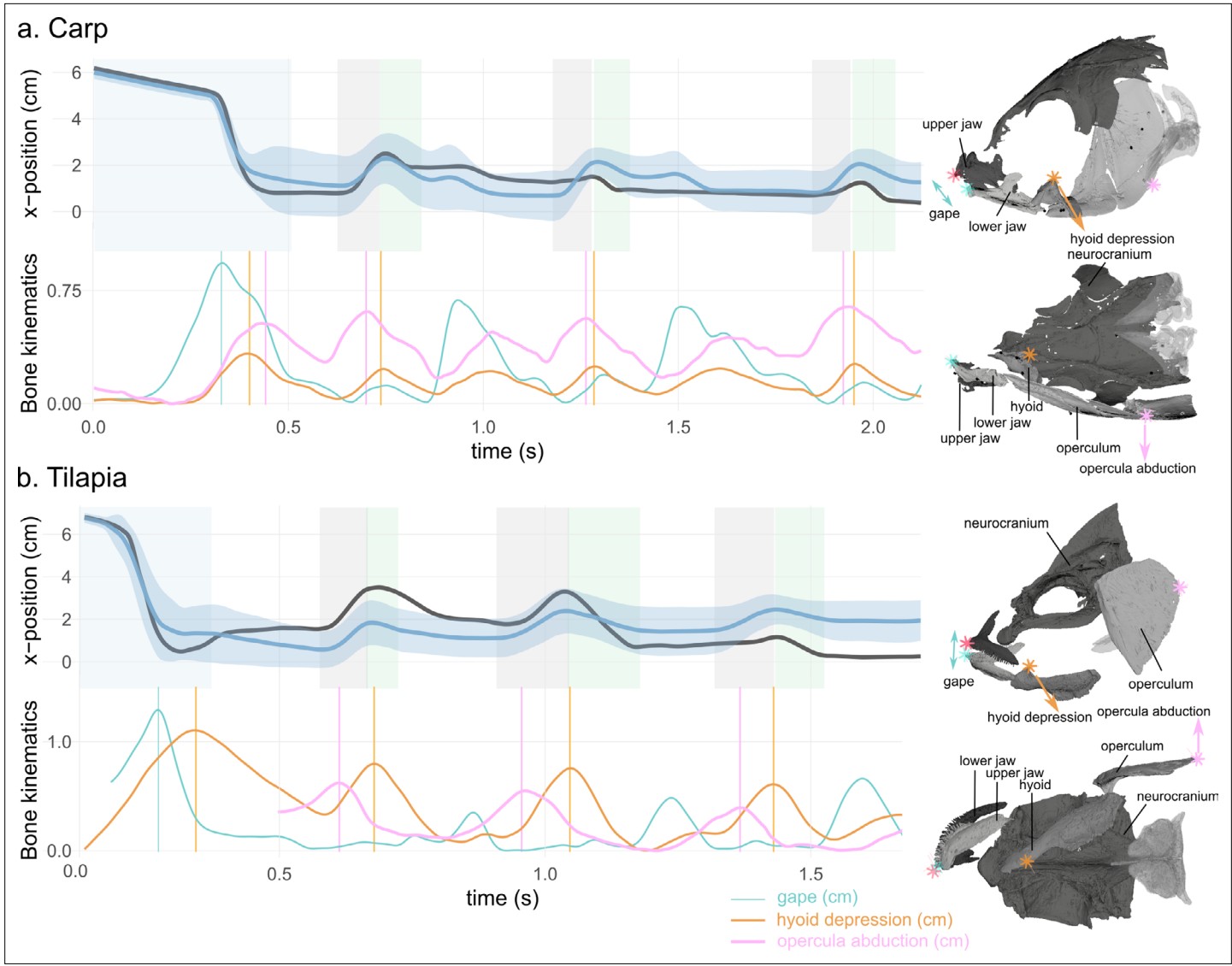

**Figure 5.** Water tracer anteroposterior trajectory, gape, hyoid depression, and opercula abduction during representative suction-feeding sequences in carp (**a**) and tilapia (**b**). The shaded bars in the x-position graphs represent theintake (blue), reverse flow (rf, grey), and backflow (bf, green) phases. The vertical lines in the bone kinematic plots represent the time of the peak gape during the intake (in blue), peak hyoid depression (in orange), and peak opercula abduction (pink) during the rfs. The sketches on the right represent a lateral and ventral view of a carp (top) and a tilapia (bottom) with the location of the locators used to calculate the gape, hyoid depression, and opercula abduction. Note that the opercula were out of the field view at the beginning of the selected tilapia trial; therefore, the opercula abduction trace starts after the intake. See *Figure 5—figure supplement 1* for the location of the implanted markers.

The online version of this article includes the following source code and figure supplement(s) for figure 5:

**Source code 1.** R code generating the graphs of *Figure 5*.

**Figure supplement 1.** Position of the implanted markers and the locators in the carp (**a–d**) and tilapia (**e–h**).

included a water outflow trough gill slits in 3D models of expanding heads (*Figure 6a*; *Thompson et al., 2018*) or a potential artifact of the hydrodynamic simplification in models using steady flow conditions in nonexpanding heads (*Figure 6b*; *Provini and Van Wassenbergh, 2018*). However, the current experimental results show that similar central jet patterns can be present in vivo. The computational fluid dynamics (CFD) models indicate that a diverging flow pattern (*Figure 1b*; *Figure 6a* at 2.5 ms) gradually transitions into a central jet by narrowing in course of the suction event. Since a narrowing of the main suction stream will increase the chance of food interception close to the esophagus entrance by the pharyngeal jaws (*Figure 1b and c*), this will enhance suction efficiency. Further

**Figure 6.** Results from previous computational fluid dynamics (CFD) studies of suction feeding in fish showing intraoral flow patterns. Anterior-to-posterior velocities (color scale) and streamlines at the mid-frontal section planes are given, with the respective 3D models of the head in the bottom-left corners. (**a**) Central jet development in a CFD model of suction feeding in shiner perch (*Cymatogaster aggregata*) from *Thompson et al., 2018* showing flow patterns at three consecutive instants during suction-feeding event (see *Figure 6—video 1*). (**b**) Model of steady flow into the mouth of a scan-based reconstruction of the head of a sunfish (*Lepomis gibbosus*) indicating approximately stagnant water (in green) close to the opercula and a jet shearing the posterior pharynx before exiting the cavity (after *Provini and Van Wassenbergh, 2018*). Scale bar, 10 mm.

The online version of this article includes the following video for figure 6:

**Figure 6—video 1.** Video showing the central jet development in a computational fluid dynamics (CFD) model of suction feeding in shiner perch (*Cymatogaster aggregata*) from *Thompson et al., 2018* showing flow patterns at three consecutive instants during suction-feeding event.
https://elifesciences.org/articles/73621/figures#fig6video1

research will be required to unravel why the central jet was more prominent in tilapia (*Figure 4b*) compared to carp (*Figure 4a*), which should be related to differences in mouth shape, buccal cavity shape, or expansion kinematics.

Secondly, the path curvature from a dorsoventral viewing perspective was significantly higher for the water tracers compared to the food tracers (*Figure 4*), indicating that inertial effects are at play. For the large prey that fill the bulk of the buccopharyngeal cavity, such inertial effects would be logical and intuitive. However, our data suggest that these mechanisms, in essence a type of filtration, are also present and effective for a relatively small pellet. A similar mechanism is hypothesized to underlie the process of crossflow filtration in filter feeders in the flows that pass close to the branchial arches (*Sanderson et al., 2016*; *Sanderson et al., 1991*), but here we show that such an inertial separation mechanism is present at the full scale of the buccopharyngeal cavity. The effect of the size, shape, and density of food on its trajectory in a flow field, as shown in our work (*Figure 3*, *Figure 3—figure supplement 1*), remains to be further quantified to fully understand the constraints of the observed filtration mechanism during the initial phase of food intake. Although flow patterns matching those of a branchial crossflow could not be identified (*Figure 1d*), the fish may not have employed this technique because the food type that was offered may not require it to be filtered in such a way.

As shown in previous work (*Olsen et al., 2019*; *Sibbing, 1982*; *Sibbing et al., 1986*; *van Meer et al., 2019*; *Weller et al., 2020*), the feeding process is far from finished after the initial capture in the buccopharyngeal cavity, especially in omnivorous species that perform food sorting and processing. We observed regular and periodical back-and-forth motions of the water tracers after this intake, which moved the food back and forth inside the buccal cavity, with the food finally ending up near the entrance of the esophagus (*Figure 2*). Rfs, directed toward the anterior part of the buccal cavity and previously referred as 'back-wash' (*Sibbing et al., 1986*), were previously observed in carp (*Callan and Sanderson, 2003*) and tilapia (*Smith and Sanderson, 2008*). In carp, anterior movement of the food has been observed using X-ray videos, after which a muscular structure packed with taste buds – the palatal organ – is hypothesized to locally bulge to clamp onto edible particles between the pharyngeal roof and floor, while small waste particles are flushed through the branchial slits (*Sibbing, 1988*; *Sibbing et al., 1986*). The anteromedial direction of the flows observed in this study during the rf phase (*Figure 3b*) seems ideal to bring food particles from both the gill rakers and from the pharyngeal region toward the palatal organ. Overall, these back-and-forth flows are clearly important in the process of sorting food items from the nonedible particles present in the ingested water, and

they potentially also prevent the gill slits from clogging. Rfs most likely involve the gill slit functioning as a flow inlet, a function that has rarely been studied (*Van Wassenbergh et al., 2016*).

Our data provide the first empirical evidence of bidirectional waterflows being synchronized with motions of the upper and lower jaws, hyoid, and opercula to create anterior-to-posterior waves during bfs and posterior-to-anterior waves during rfs. The anterior-to-posterior wave of head expansion and compression is already well known in fish (*Gibb and Ferry-Graham, 2005*; *Lauder, 1985*; *Muller and Osse, 1984*; *Van Wassenbergh et al., 2016*). This is interpreted as a mechanism to both ingest and dynamically accommodate a volume of water as it moves anteroposteriorly through the buccal cavity. For both species studied, this classical anterior-to-posterior wave was also observed during the consecutive periodic bf motions. Yet, the associated waterflow was not purely anterior-to-posteriorly directed: the water was also guided toward the ventral part of the buccal cavity (*Figure 3*) by the hyoid depression, and it deviated toward the gill covers (*Figure 4*) via the abduction of the opercula (*Figure 5*). During the backward motions of the water, we never observed any particles entering the digestive tract, thus consolidating our hypothesis that no water enters inside the esophagus. A wave in the opposite direction was synchronized with the rfs, as hypothesized earlier (*Sibbing et al., 1986*; *Van Wassenbergh et al., 2016*). The stereotypical bf and rf alternations eventually lead the food to the centerline of the buccal cavity and closer to the pharyngeal jaws, where it will be actively transported toward the digestive tract (*Sibbing, 1982*).

In conclusion, our study highlights the biomechanical complexity of the processes occurring inside the mouth cavity during aquatic feeding in fishes. We showed how high-velocity suction flow patterns during the initial food intake phase can have an effect on the food's initial deposit site, and how centralized suction streams may help to direct food toward the esophagus. The omnivorous species studied here during benthic feeding followed this phase by repeated, hydrodynamically driven repositionings as part of their food sorting and tasting abilities. Future research using our X-ray particle-tracking technique could further reveal how these processes, and their intraoral hydrodynamic patterns, may vary between feeding modes (e.g., ram-feeders ingesting larger volumes of water), between animals with varying head morphology and size, etc. Such studies would further advance our currently limited insight into the optically invisible processes of aquatic feeding in vertebrates.

## Materials and methods
### Animal care and surgical procedures

Two carp (*C. carpio*), carp 01 (mass = 341 g, total length = 25.94 cm) and carp 02 (mass = 304.5 g, total length = 20.42 cm), purchased from a pond shop (De Hof-Leverancier, Ekeren, Belgium), and two Nile tilapia (*O. niloticus*), tilapia 01 (mass = 488.4 g, total length = 29.4 cm) and tilapia 02 (mass = 489.0 g, total length = 30.3 cm), from a farm (Til-Aqua, Someren, The Netherlands), were housed in two large water tanks at the University of Antwerp and provided with food ad libitum.

We reduced the number of experimental animals to a minimum for ethical considerations. As quantifying individual variability would require a higher number of individuals per species, it was not a goal of our study; therefore, intraspecific variation is not covered sufficiently by our analysis to allow generalizations within each species. Additionally, the limited number of species studied does not allow us to generalize our findings to other adult, omnivorous, teleost fish. Even if the number of analyzed individuals was limited to two per species, which may limit the interpretation of the data in the context of the STRANGE framework, the total number of individuals across both species (four individuals), the number of trials (carp: seven trials; tilapia: six trials), and the consistency of the results imply that the mainly qualitative conclusions on overall patterns in waterflow and skeletal motion are well supported.

Prior to videography, the fish were anesthetized with 50 mg/L MS222 (ethyl 3-aminobenzoate methanesulfonate) and implanted with 0.35-mm-diameter beads made of a tin and silver alloy. Using hypodermic needles, four markers were implanted in the neurocranium, and three on the upper jaw, lower jaw, hyoid, and left operculum (*Figure 5—figure supplement 1*). The left suspensorium, left cleithrum, left fin, and two of the branchial arches were implanted with one or two markers for reference. We also implanted 4-mm-diameter food pellets with three 0.45-mm-diameter markers. After the surgery, the fish were kept under observation until full recovery. The experiments were ethically approved by the University of Antwerp (ECD-2017-22).

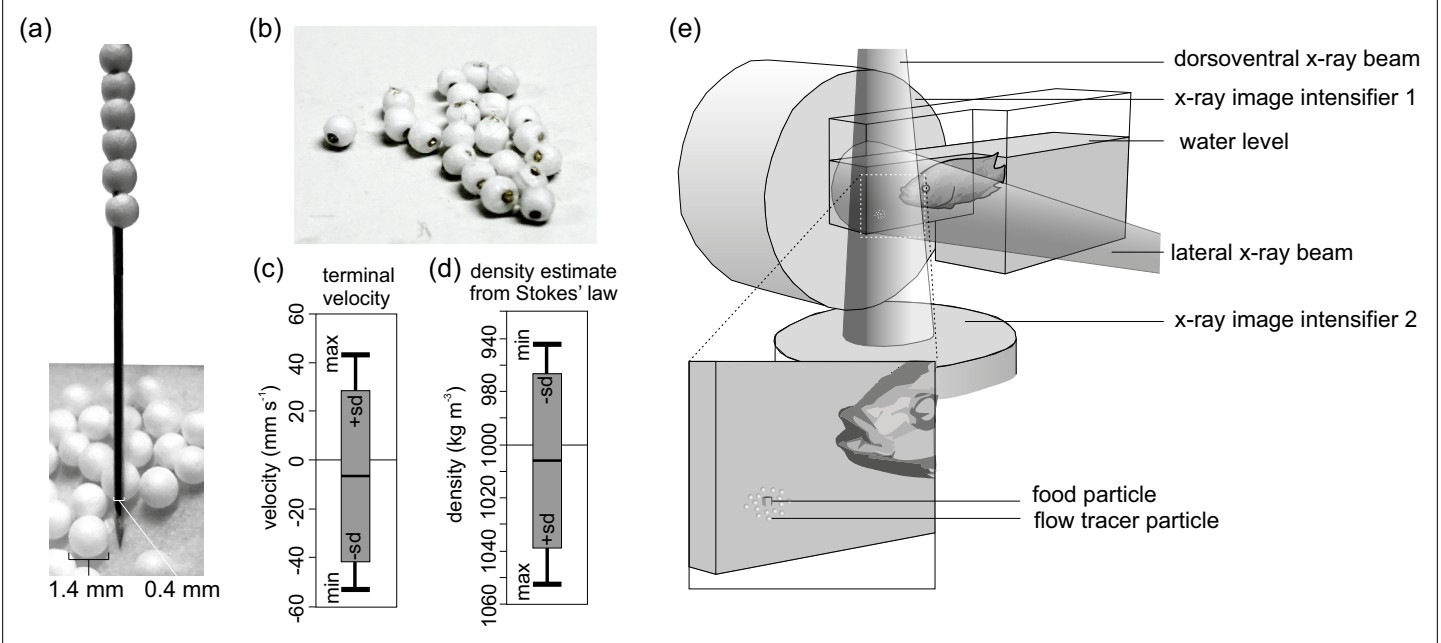

**Figure 7.** Tracer fabrication for X-ray particle tracking and the experimental design. (**a**) Construction of the flow tracer particles by piercing 1.4-mm-diameter expanded polystyrene (EPS) spheres on a 0.4-mm-diameter, Vaseline-coated brass rod with a sharpened tip. (**b**) Flow tracers after cutting the rod. (**c**) Terminal velocity test results of the EPS brass flow tracers released in water (positive velocities: rising; negative velocities: sinking), and (**d**) the corresponding estimates of density based on Stokes' law (see Materials and methods for details) (N = 19). (**e**) Experimental design indicating X-ray beam, image intensifier orientation, and placement of the food surrounded by flow tracer particles at the bottom of the narrow extrusion of the aquarium.

The online version of this article includes the following source code for figure 7:

**Source code 1.** R code generating the graphs of *Figure 7*.

## Water tracers

### Design of the water tracers

The water tracers had to fulfill two typically incompatible requirements: (1) being radio-opaque to be trackable on the X-ray videos and (2) being small enough and neutrally buoyant to passively follow the water trajectory as closely as possible. An easy-to-execute and cost-efficient procedure was developed to make spherical waterflow tracers of minimal size to be used with biplanar high-speed X-ray video setups. Like the existing tracer designs (*Drake et al., 2011*; *Seeger et al., 2001*), we surrounded a radio-opaque metal with a closed-cell foam. The foam compensates for the inevitably high weight of the metal so that the overall density of the tracer particle approximates the density of water.

EPS foam spheres with diameters between 1 and 2 mm were purchased (Rovul, Middelstum, The Netherlands) and sieved to retain only those with a diameter of 1.4 mm. To do so, the foam spheres that were loosely stuck in between the mesh holes of a precision test sieve were extracted. Next, 0.4-mm-diameter brass rods (Albion Alloys, Bournemouth, UK) were cut into approximately 0.1 m pieces in length, and their tips were sharpened by holding the rod at a sharp angle against a grinding stone rotating on a Proxxon MF70 micro-milling machine (Proxxon GmbH, Föhren, Germany). Next, the sharp tips were dipped in petroleum jelly to reduce friction. A foam sphere was then gently immobilized between two fingertips while being pierced (as close as possible through its center) by the sharp tip, which was gently rotated by the fingertips of the other hand (*Figure 7a*). The one-by-one pierced EPS spheres were slid to the back of the rod. Finally, when the rod was filled with a chain of foam spheres, each sphere was clipped close to the foam sphere edges using side-cutting pliers (*Figure 7b*).

Although these particles are large compared to traditional (non-X-ray-based) particles used in particle-tracking velocimetry, the volume of our 1.4-mm-diameter tracers is more than five times smaller than the smallest tracers used so far for X-ray-based particle tracking (*Seeger et al., 2001*; 2 × 2 × 2 mm cubes), which in turn were already considerably smaller than the 8-mm-diameter spheres

for which a manufacturing protocol has been described in detail (*Drake et al., 2011*). In relation to the size of the buccopharyngeal cavity of the individual studied – roughly 65 × 50 × 40 mm – these tracers are sufficiently small to resolve the large-scale flow structures.

## Estimation of the food tracer volume and density

To estimate the volume of the food items used in the experiment, we laser-scanned 10 food items (Faro Laser ScanArm V2 system). We obtained a volume (mean ± SD; N = 10) of $24 ± 2\ mm^3$.

The food items implanted with radio-opaque markers were weighted to obtain their dry mass and were soaked in water for 15 min to estimate their wet mass. We obtained a density (mean ± SD; N = 10) of $1845 ± 170\ kg/m^3$.

## Test of the water tracer density

Nineteen water tracers held by forceps in the middle of a water column were released and filmed at 50 frames per second next to a scale bar to extract their trajectory using XMALab software version 1.5.1 (https://www.xromm.org/xmalab/; *Knörlein et al., 2016*). We measured the terminal velocity ($V_{terminal}$) of the tracer (*Figure 2c*) and calculated the water tracer density ($\rho_{tracer}$) using Stokes' law (*Equation 1*):

$$\rho_{tracer} = \frac{18\ \mu_{water} \times V_{terminal}}{g \times d^2} + \rho_{water} \qquad (1)$$

where $\mu_{water}$ corresponds to the dynamic viscosity of water – that is, 1.002 mPa·s at 20°C (*Cooper and Dooley, 1993*) – g to the gravitational acceleration of 9.81 $m/s^2$, d to the diameter of the tracer including both the nickel rod and the foam sphere – that is, 1.4 mm –, and $\rho_{water}$ to the water density of 1000 $kg/m^3$. The mean (± SD) density of the produced particles from the test sample (N = 19) was 994 ± 33 $kg/m^3$ (*Figure 7d*). Since only negatively buoyant particles were presented at the bottom of the aquarium surrounding the food (*Figure 7e*), the mean particle density was 1031 ± 14 $kg/m^3$, with a maximum of 1049 $kg/m^3$ (N = 11).

## Computational fluid dynamics

To evaluate the performance of our particles for tracking waterflows during suction feeding, numerical simulations were performed using the CFD technique. A simple model of suction into a cylindrical cavity (*Figure 8a*, *Figure 8—video 1*), with a cavity size (length 40 mm, radius 10 mm) and suction pressure magnitude (sinusoidal profile reaching - 2 kPa after 20 ms) that are realistic for adult fishes,

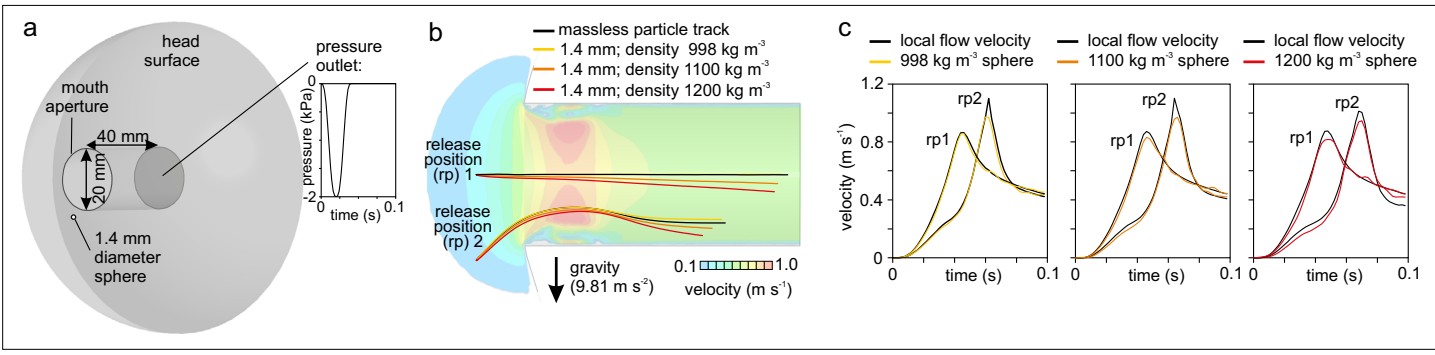

**Figure 8.** Numerical modeling assessment of tracking performance of suction-feeding waterflows. (**a**) Cylindrical mouth cavity inside a hemispherical head generating suction trough a transient pressure-outlet-type boundary condition assigned to the back of the cylinder. (**b**) Path lines of the water (black lines) released from two points in front of the mouth aperture (rp 1 and 2) in comparison with trajectories from 1.4-mm-diameter spheres with density equal to the water (998 $kg/m^3$; yellow) and two cases with negative buoyancy (1100 $kg/m^3$, orange; 1200 $kg/m^3$, red). Note that the direction of gravity is downward. Flow velocity color maps are shown for the simulation time of 0.06 s (see *Figure 8—video 1*). (**c**) Comparison of the velocity between the particles (colored lines) and waterflow at the current position of the particles in a flow simulation in the absence of the particle (black lines).

The online version of this article includes the following video for figure 8:

**Figure 8—video 1.** Video of the computational fluid dynamics (CFD) simulation testing the influence of the tracer density on their trajectory during the intake.

https://elifesciences.org/articles/73621/figures#fig8video1

was used in laboratory studies. These simulations were performed in software Ansys 2019 R1 (Ansys Inc, Canonsburg, USA) using the CFD solver Ansys Fluent. The cylindrical mouth cavity was extruded out of a hemispherical "head", which, in turn, together with a 0.7 mm radius sphere were subtracted from a large hemisphere with 0.6 m radius in Ansys DesignModeler. The resulting volume, that is, the flow domain, was meshed with about 2 million tetrahedral cells (mesh edge size of 0.3 mm at the cylinder; growth rate of 1.1) using Ansys Meshing. A simulation with a less-refined mesh of about 1 million tetrahedra (mesh edge size 0.5 mm) showing nearly identical flow patterns confirmed that the 2 million cell mesh was fine enough. In Ansys Fluent, gravity was enabled (9.81 $m/s^2$) in a transient, laminar flow model using the 6 DOF dynamic mesh functionality to update the position of the spherical particle to the next time step (800 time steps of 0.125 ms solved with 40 iterations per time step). The inertial properties of spheres filled with uniformly dens material, as well as the hydrostatic lift force, were provided by user-defined functions. A second user-defined function set the profile of the pressure drop at the interior base plane of the cylinder (*Figure 8a*). The dynamic mesh updating involved both smoothing (spring constant factor 0; Laplace node relaxation factor 0.5) and local cell remeshing at each time step. The default mathematical solver settings for Fluent were used. We ran simulations for a case without a flow-tracing sphere, for two simulations with a sphere with the same density as water (998 $kg/m^3$) at two initial positions (*Figure 8b*), and similarly for spheres with densities of 1100 and 1200 $kg/m^3$.

The CFD results reassure that the finite size (1.4 mm diameter) and density (up to about 1050 $kg/m^3$) of the current sample of particles (*Figure 8*) do not hinder a realistic assessment of the suction flows by fish. A small deviation of the path in the direction of gravity can be expected (*Figure 8b*), but this should be smaller than 1 mm even for the heaviest particles of 1050 $kg/m^3$. Lag during flow acceleration and overshoot during deceleration for the slightly negatively buoyant particles were relatively small (*Figure 8c*) and therefore acceptable to describe general patterns of flow during suction feeding.

## Biplanar X-ray videography

The four implanted fish were recorded during their feeding on a marked food item surrounded by about 15 radio-opaque water tracers. They were filmed at a resolution of 2048 × 2048 pixels and at a speed of 750 frames per second using the 3-D DYnamic MOrphology using X-rays (3D²YMOX) set-up at the University of Antwerp (*Sanctorum et al., 2019*). The system consists of two X-ray videography systems, each composed of a Photron FastCam Mini WX50-32GB camera (Photron USA, Inc, San Diego, CA) mounted on a Philips Imagica HC 38 cm image intensifier. The two conical X-ray beams were set to create a lateral and a dorsoventral view (*Figure 7e*) using the image intensifier's smallest field-of-view setting for the carp and the medium field-of-view setting for the tilapia, emitting at tube voltages and currents of 75 kV and 80 mA and 95 kV and 85 mA, respectively (*Figure 7e*). More technical details and the performance tests of the 3D²YMOX system can be found in a previous publication (*Sanctorum et al., 2019*).

We followed recommendations regarding distortion correction and space calibration during the experiment (*Brainerd et al., 2010*; *Gatesy et al., 2010*) and used a perforated steel sheet with a precise hole size and spacing for distortion correction and a calibration object for determining camera 3D-view characteristics. We recorded 73 sequences of suction feeding of around 5 s in duration and selected 13 sequences corresponding to successful attempts (e.g., fish head in the field of view during the entire sequence, water tracers successfully ingested with the food item) balanced among individuals. The fish were euthanized with MS222 and scanned using micro-computed tomography (μCT), with the implanted markers intact (μCT scans are available at the Royal Belgian Institute of Natural Sciences; 110 kV, 0.5 mA, and a resolution of 64.94 μm on each axis). We used Avizo (version 6.3; FEI Visualization Sciences Group) to reconstruct bone models and the implanted markers from the μCT scans.

## X-ray reconstruction of the moving morphology

Using the validated XMALab software version 1.5.1 (https://www.xromm.org/xmalab/; *Knörlein et al., 2016*), we undistorted the video images, calibrated the view of the X-ray cameras, and tracked the three types of markers (bone markers, food and water tracers) on each frame of the selected videos.

### Rigid bodies

To reconstruct the 3D motion of the implanted bones, we followed the XROMM workflow; we imported the 3D bone models into Maya (version 2018; Autodesk) and extracted CT marker coordinates. We generated the rigid-body transformations in XMALab and imported them into Autodesk Maya to animate the bones. We used scientific rotoscoping (*Gatesy et al., 2010*) to position rigid bodies with less than three markers (in case of losses after surgery or when markers were out of view).

### Coordinate system definition

To obtain a common framework between trials, we created an anatomical coordinate system (ACS) attached to the neurocranium. The origin of this ACS was offset to be positioned close to the location of the pharyngeal jaws as an approximation of the esophagus entrance location (midsagittal plane, caudal tip of the chewing pad). The pharyngeal jaws are known to move during the food capture and transport (e.g., *Claes and De Vree, 1991*; *Sibbing et al., 1986*), but we did not reconstruct their motions in this study. Their static position was used as an estimation of the esophagus entrance location. The x-axis was parallel to the plate of the pharyngeal process, pointing toward the entrance of the buccal cavity, and was positive anteriorly. The y-axis corresponded to the midsagittal axis, perpendicular to the x-axis, and was positive dorsally. The z-axis vector (positive right) was calculated by crossing the proximal x-axis with the y-axis (*Figure 2*).

### Food and water tracers

Among the 73 trials recorded, the entire buccal cavity and tracers were not always visible throughout the entire sequence; therefore, we focused our analysis on seven trials among the two individuals of carp and six trials among the two individuals of tilapia. For these trials, we imported the 3D trajectories of the food and water tracers into Maya (version 2018; Autodesk). We computed the relative 3D trajectory and relative 3D velocity in the coordinate system previously described for the water and food tracers. We computed the mean and standard deviations of these parameters among trials within each species (*Figure 2*, *Figure 2—video 1*, *Figure 2—video 2*).

### Phases of suction feeding

To divide the suction-feeding sequences into phases, we used the anteroposterior trajectory and velocity of the water tracers (*Figure 2—source code 2*). We excluded the stagnant and slow-moving parts of the path lines by discarding velocities between −4.5 and 2 cm/s from the analysis.

The 3D trajectory of all the water tracers during each suction-feeding sequence provided a visualization of the water tracer paths for the different views of a representative trial and of all the trials of a given species during the intake (*Figure 3a*), the rf phase (*Figure 3b*), and the bf phase (*Figure 3c*).

We registered the number of successful suction-feeding sequences among all the recorded trials (N = 37 in carp, N = 36 in tilapia). For this set of data, we noted the number of food and water tracers reaching the entrance of the esophagus that were effectively trapped in the digestive tube when the tracer was at the level of the pharyngeal jaws. We also measured the number of water tracer oscillations (backward and forward motions) before the food tracer reached the area of the pharyngeal jaw's axis.

In addition, we compared the global distance traveled by the food tracers and the water tracers. We computed the relative contribution of each component of the trajectory during the intake, the rf phases, and bf the phases among the studied trials (*Figure 2—figure supplement 1*, *Figure 2—figure supplement 1—source code 1*).

### **Path line**

To quantify the deviation of the water tracers relative to the food tracers, we calculated the curvature of a given tracer based on three consecutive frames in the dorsoventral plane (x, z). To do so, we standardized the position of the tracers on the anteroposterior axis by calculating the x-trajectory percentage for each tracer, then we computed the curvature values over these intervals for the food tracers and the water tracers. As the distribution was not normal, we performed a Kruskal–Wallis rank-sum test to evaluate if the differences in the maximum curvature per trajectory of each interval were significantly different between the water tracers and the food tracers (*Figure 4*, *Figure 4—source code 1*).

We plotted the tracer coordinates in a dorsoventral plane (tx, tz) and assigned a color based on their location at 25% of the intake phase duration for all the tracers of all the trials of each individual (*Figure 4—figure supplement 1*, *Figure 4—figure supplement 1—source code 1*). This provided information on the distribution of the tracers according to the moment they were ingested. We tested the correlation between the anteroposterior location at 25% of the intake phase duration (tx) and the lateromedial location at the end of the intake (tz) (Kendall's correlation test).

To explore the relationship between the initial position of the tracers at the beginning of the first intake to their final position at the end of the first intake, we plotted the tracer coordinates (tx, tz) and assigned a color based on their initial location for all the tracers of all the trials of each individual (*Figure 4—figure supplement 2*, *Figure 4—figure supplement 2—source code 1*). We tested the correlation between the lateromedial location at the beginning of the intake phase (tz) and the lateromedial location at the end of the intake (tz) (Kendall's correlation test).

## Skeletal kinematics

The exclusion criteria were even more drastic for this part of the analysis as we needed not only the buccal cavity and tracers visible throughout the entire sequence, but also the bone markers. We had to withdraw the data for one individual of tilapia, consistently out of the field of view at the end of the first intake. Therefore, we ended up with two individuals of carp and seven trials, one individual of tilapia and three trials.

We exported the 3D trajectories of additional locators we positioned on the anterior tip of the upper jaw, lower jaw, and hyoid, as well as on the posterior tip of the operculum on the animated Maya scenes (*Figure 5—figure supplement 1*). This allowed us to compute the gape, hyoid depression, and opercula abduction for six carp sequences and three tilapia sequences. We extracted the time when the peaks of each variable were reached during the intake and the rfs for each analyzed sequence.

## Acknowledgements

We would thank Peter Aerts for access to the facility at the University of Antwerp, Jonathan Brecko for the CT-scan acquisition at the Royal Belgian Institute of Natural Science, as well as Anthony Herrel for helpful discussions. This work was funded by the Project-ANR-16-ACHN-0006 at the MNHN-UMR 7179 Muséum national d'Histoire naturelle and supported by the University of Antwerp grant BOF KF110000 (DynXlab).

## Additional information

### Funding

| Funder | Grant reference number | Author |
| --- | --- | --- |
| Agence Nationale de la Recherche | ANR-16-ACHN-0006 | Sam Van Wassenbergh |

The funders had no role in study design, data collection and interpretation, or the decision to submit the work for publication.

### Author contributions

Pauline Provini, Conceptualization, Data curation, Formal analysis, Investigation, Methodology, Validation, Visualization, Writing - original draft, Writing - review and editing; Alexandre Brunet, Andréa Filippo, Formal analysis, Writing - review and editing; Sam Van Wassenbergh, Conceptualization, Funding acquisition, Investigation, Methodology, Project administration, Supervision, Writing - original draft, Writing - review and editing

### Author ORCIDs

Pauline Provini  http://orcid.org/0000-0002-9374-1291
Sam Van Wassenbergh  http://orcid.org/0000-0001-5746-4621

## Ethics

This study was performed in strict accordance with the European recommendations of animal experimentation. All of the animals were handled according to approved institutional animal care and were ethically approved by the University of Antwerp (ECD-2017-22). All surgery was performed under Ethyl 3-aminobenzoate methanesulfonate anesthesia, and every effort was made to minimize suffering.

## Decision letter and Author response

Decision letter https://doi.org/10.7554/eLife.73621.sa1
Author response https://doi.org/10.7554/eLife.73621.sa2

---

## Additional files

### Supplementary files
• Transparent reporting form

### Data availability
All data analysed during this study are included in the manuscript and supporting files.

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
