## [Editor Report]

How do fish suck food underwater? Using new artificial food particles that are radio-opaque and naturally buoyant, Provini et al. imaged the roller-coaster ride that food particles make being sucked in from outside to inside the fish using 3D stereo high-speed fluoroscopy. The recordings show fishes to have an intriguing ability to generate flows that center the food particles as they enter the muscular tube that carries them from the mouth to the stomach. Remarkably, the flow patterns in the mouth that accomplish this seem to differ between the two species of fish studied, although samples sizes are very small at present and results should be interpreted cautiously. These new insights will be of interest to biologists working on suction-feeding mechanisms ranging from millimeter-sized carnivorous water plants, tadpoles, and fish larvae, to large fish and marine mammals, and even gigantic whales. Bioinspired engineers designing rapid underwater suction apparatuses may benefit from harnessing the new insights to elegantly center items of interest.

---

## [Decision Letter]

**Decision letter after peer review:**

Thank you for submitting your article "in vivo intraoral waterflow quantification reveals hidden mechanisms of suction feeding in fish" for consideration by *eLife*. Your article has been reviewed by three peer reviewers, including David Lentink as Reviewing Editor and Reviewer #1, and the evaluation has been overseen by Christian Rutz as the Senior Editor.

The reviewers have discussed their reviews with one another, and the Reviewing Editor has drafted this decision letter to help you prepare a revised submission. Please carefully address the comments and suggestions, mark all changes in the revised manuscript in coloured font, and provide a point-by-point response. This will help the Reviewing Editor evaluate your revision. Please add line numbers in case another round of review is required.

Essential revisions:

Overall, the editors and reviewers all greatly appreciate the method used, the experiments performed, and the data visualization presented. This has persuaded us to give the authors the opportunity to resolve several weaknesses in a revised manuscript.

– High-level feedback based on editorial discussion

While this work on fish suction feeding is impressive and informative, it is currently too narrowly focused for a broad interest journal like *eLife*. Hence, we believe the manuscript would be much improved if it succinctly addressed the following points.

(1) A broader introduction on suction feeding in its ecological and evolutionary context is needed, covering a wider range of suction feeding studies, from tadpoles, fish larvae and microscopic carnivorous waterplant suction feeding apparatuses up to those of the largest marine vertebrates (e.g., whales). The point needs to be made more compellingly that, while suction feeding has been studied across scales externally to the degree possible, the internal fluid mechanics remain poorly understood at any of those scales due to the experimental challenges of imaging flow at high-speed internally. This requires a paragraph in the Introduction, to enable the general reader to understand why this research matters and what challenge was solved for which reason; especially, if the evolutionary context of "why suction feeding matters" is given in the broadest sense. For this the following review article may be helpful: Deban, Stephen M., Roi Holzman, and Ulrike K. Müller. "Suction feeding by small organisms: performance limits in larval vertebrates and carnivorous plants." Integrative and Comparative Biology 60.4 (2020): 852-863.

(2) With this broader background in place, the authors have a solid jump-off point at the end of the Introduction (before they describe the methods used) to introduce the model species selected. This is likely based on factors such as their size matching the optimal bi-planar (stereo) fluoroscopy setup imaging volume.

(3) The text about the potential impact of the method itself in the last paragraph of the Discussion can stay (although its readability should be improved), but it isn't an effective closing paragraph for a broad audience. In the Discussion, it would be good to conclude with a paragraph with a broader perspective, one that connects to the introductory biological context added under (1). Currently the last paragraph is very dense and technical, written for a specialist audience. To better serve the broad readership of *eLife*, it would help to translate the biological implications to movement ecology or evolutionary biology or any other biological domains the authors believe will benefit from the new scientific insights. This section does not need to be speculative; it can be strongly connected to the actual findings, for example, on how they could inform other studies in which suction feeding is a key factor. For example, an effective broader impact closeout could include discussion of any scaling laws derived or confirmed based on the research, helping explain how these findings could advance our understanding of suction feeding across a wider range of species. Or, it could be a very specific and focused paragraph on the mechanism for the species tested written in a fashion that makes it more broadly accessible, with associated clear schematics connecting internal and external perspectives: a helicopter perspective completing our understanding of suction feeding (one that could go into a biology textbook). In conclusion, we encourage the authors to write a closing paragraph that is scientifically rigorous while serving a broader audience beyond the small suction feeding community. Some minor speculation is acceptable if the wording communicates this.

– Actionable feedback summary from the peer review outcomes

(4) Please clarify the number of species, individuals and trials for which data have been processed and integrated in the figure captions, Methods, Results and Discussion. The number of individuals per species (n = 2) is very small and this has been brought up as a concern. Another concern is that the number of individuals and trials analyzed seems to be variable throughout the manuscript, depending on the particular analyses made. To address this, we would like to see all data processed for all individuals and species reported in the manuscript. In the Methods, the rationale for using just 2 individuals, as compared to the more typical 3-4 as a minimum, is missing. In the Discussion, a few sentences should clarify how the low number of individuals may limit the interpretation and generalisability of the data. Given the total number of individuals across both species is 4, the overall conclusions supported across both species and both individuals are considered strongest. Finally, please keep in mind that the calculation of a biologically meaningful average and standard deviation representative for a species requires a minimum of three individuals. The major limitations remain (after processing all trials for all individuals across the two species to resolve the minor limitations) and impact the ability to conduct statistical analyses, inferences and conclusions that are biologically meaningful. Yet, this context is not sufficiently integrated in the presentation of the experimental design and discussion of the results.

(5) The tracking of particles into the mouth should provide qualitative data that can highlight the hydrodynamics of the flows, but the speed of intra-oral flows is not fully examined. The manuscript is missing a quantitative estimate of the volume of water that passes near the esophagus, and of the kinematics of bone movements and how they correspond to fluid flows. Therefore, we would like to see quantitative mapping of the relationship between the distance from the mouth at the beginning of the strike to where the particle ends, and other quantitative measures. The authors also mention efficiency but make no attempt to quantify it. For the reviewers, it is the quantitative insight into these and other suction feeding parameters that has the potential to make this article particularly impactful.

(6) The beautifully illustrated tracks are a strength of the manuscript, but also mask some mechanistic details we would like to see dissected. The tracks in the current plot format confound space and time, meaning that the particles potentially started at different distances under different angles from the mouth and may have also started to move substantially at different times. We therefore believe additional figures are needed to quantitatively analyze this. Quantification will also enable the authors to make statements such as "food items that are initially somewhere close to the mouth…" as specific as possible based on the dataset.

(7) We are curious if the suction effort of the fish may have varied across the trials (videos). Such variation has been shown to be primal in determining the speed of the external flows, hence we believe it is worthy of inclusion in the analyses.

(8) The manuscript somewhat oversells the study's novelty; please try to better delineate the contribution. (i) Briefly clarify the exact contribution of the new method in the context of existing literature; what existed and has been done, which aspects are added / new etc. For example, the water markers seem to represent a modified design of previously reported particle and food item tracking approaches. Neither the XROMM technique nor particle tracking nor neutrally buoyant particle tracking are entirely new; for example, tracking marked prey items (with radio opaque markers) into the mouth and esophagus using XROMM has been done previously. Both particle tracking and particle image correlation are established for a wide range of applications. XROMM is also established for 3D imaging and tracking in the body. The new advances should be presented in the context of the relevant experimental literature. (ii) Discuss previous CFD studies more extensively, and how the limitations of those studies precluded the insights achieved by the present study. Considering these internal flows were modeled extensively by the authors (others), there is a model and theoretical framework that predicts how the internal flow behaves. This rich CFD and literature context is missing in the presentation of Figure 1, the Introduction and Discussion.

(9) Please discuss the tracking particle physical properties consistently and correctly: there is confusion about the particles being neutrally buoyant versus (page 16) negatively buoyant; this should be clarified throughout. The PIV literature provides a useful tracking error framework for relating the effect of particle size and density ratio to the ability of particles to track the flow. These established error analyses should be integrated into the Methods section in which the method is outlined. This should include estimated limitations due to particle size and buoyancy imperfections limiting the ability of the particles to trace out the flow pattern of interest. With this context the reader can fully comprehend the significant advance the current study achieves, within the acceptable limitations given the challenges involved in this tracking approach. The technical discussion paragraph could comment on how future tracing particles could/should be improved.

---

## [Author Response]

Essential revisions:Overall, the editors and reviewers all greatly appreciate the method used, the experiments performed, and the data visualization presented. This has persuaded us to give the authors the opportunity to resolve several weaknesses in a revised manuscript.

The authors want to thank the editors and reviewers for the valuable feedback they provided and for the opportunity to submit a revised version of the manuscript.

– High-level feedback based on editorial discussionWhile this work on fish suction feeding is impressive and informative, it is currently too narrowly focused for a broad interest journal like eLife. Hence, we believe the manuscript would be much improved if it succinctly addressed the following points.(1) A broader introduction on suction feeding in its ecological and evolutionary context is needed, covering a wider range of suction feeding studies, from tadpoles, fish larvae and microscopic carnivorous waterplant suction feeding apparatuses up to those of the largest marine vertebrates (e.g., whales). The point needs to be made more compellingly that, while suction feeding has been studied across scales externally to the degree possible, the internal fluid mechanics remain poorly understood at any of those scales due to the experimental challenges of imaging flow at high-speed internally. This requires a paragraph in the Introduction, to enable the general reader to understand why this research matters and what challenge was solved for which reason; especially, if the evolutionary context of "why suction feeding matters" is given in the broadest sense. For this the following review article may be helpful: Deban, Stephen M., Roi Holzman, and Ulrike K. Müller. "Suction feeding by small organisms: performance limits in larval vertebrates and carnivorous plants." Integrative and Comparative Biology 60.4 (2020): 852-863.

We have added a paragraph in the introduction to broaden the scope of our work. We now mention other aquatic organisms, from larval fish to whales and carnivorous plants.

We also cited the work mentioned in your comment as well as the following references:

- Herrel, A., Van Wassenbergh, S. and Aerts, P. (2012). Biomechanical studies of food and diet selection. In: eLS, pp. 1-9. Chichester: Wiley.

- Alexander, R. McN. (1969). Mechanics of the feeding action of a cyprinid fish. J. Zool., Lond. 159, 1-15.

- Drost, M. R., Muller, M. and Osse, J. 1988 A quantitative hydrodynamical model of suction feeding in larval fishes: the role of frictional forces. Proc. R. Soc. B 234, 263–281. doi:10.1098/rspb.1988.0048

- Deban SM, Olson WM. 2002. Suction feeding by a tiny predatory tadpole. Nature 420:41–42.

Werth AJ. 2004. Functional morphology of the sperm whale (Physeter macrocephalus) tongue, with reference to suction feeding. Aquat Mamm 30:405–418.

- Müller, U.K., Berg, O., Schwaner, J.M., Brown, M.D., Li, G., Voesenek, C.J., van Leeuwen, J.L. (2020). Bladderworts, the smallest known suction feeders, generate inertia-dominated flows to capture prey. New Phytologist 228, 586-595.

- Wainwright, P.C., McGee, M.D., Longo, S.J. and Hernandez, L.P. (2015) Origins, innovations, and diversification of suction feeding in vertebrates. Integr. Comp. Biol. 55, 134-145.

- Werth, A. J. (2004). Functional morphology of the sperm whale (Physeter macrocephalus) tongue, with reference to suction feeding. Aquatic Mammals 30, 405–418

(2) With this broader background in place, the authors have a solid jump-off point at the end of the Introduction (before they describe the methods used) to introduce the model species selected. This is likely based on factors such as their size matching the optimal bi-planar (stereo) fluoroscopy setup imaging volume.

We added a sentence at the end of the introduction to present the model species selected:

“These species were chosen from the large diversity of suction feeders because their head size as adult is small enough to fit the imaging volume of the biplanar X-ray setup, yet large enough to avoid hindrance from an intake of tracer particles for waterflow inside the buccal cavity using X-rays.”

(3) The text about the potential impact of the method itself in the last paragraph of the Discussion can stay (although its readability should be improved), but it isn't an effective closing paragraph for a broad audience.

We streamlined the last paragraph of the Discussion of the previous version of the manuscript.

In the Discussion, it would be good to conclude with a paragraph with a broader perspective, one that connects to the introductory biological context added under (1). Currently the last paragraph is very dense and technical, written for a specialist audience. To better serve the broad readership of eLife, it would help to translate the biological implications to movement ecology or evolutionary biology or any other biological domains the authors believe will benefit from the new scientific insights. This section does not need to be speculative; it can be strongly connected to the actual findings, for example, on how they could inform other studies in which suction feeding is a key factor. For example, an effective broader impact closeout could include discussion of any scaling laws derived or confirmed based on the research, helping explain how these findings could advance our understanding of suction feeding across a wider range of species. Or, it could be a very specific and focused paragraph on the mechanism for the species tested written in a fashion that makes it more broadly accessible, with associated clear schematics connecting internal and external perspectives: a helicopter perspective completing our understanding of suction feeding (one that could go into a biology textbook). In conclusion, we encourage the authors to write a closing paragraph that is scientifically rigorous while serving a broader audience beyond the small suction feeding community. Some minor speculation is acceptable if the wording communicates this.

We added a concluding paragraph that (1) summarizes the biological implications of our work, and (2) provides a future perspective of research following up on our work. We feel that it should be accessible to a broad audience. The newly discovered flow type was highlighted. It has the potential to become textbook knowledge on the basic functioning of suction feeding.

– Actionable feedback summary from the peer review outcomes(4) Please clarify the number of species, individuals and trials for which data have been processed and integrated in the figure captions, Methods, Results and Discussion. The number of individuals per species (n = 2) is very small and this has been brought up as a concern.

We have clarified the number of species, individuals and trials for which data have been processed and integrated in the figure captions, Methods, Results and Discussion.

Another concern is that the number of individuals and trials analyzed seems to be variable throughout the manuscript, depending on the particular analyses made. To address this, we would like to see all data processed for all individuals and species reported in the manuscript.

The number of individuals and trials were consistent throughout the hydrodynamic analysis (carp: 2 individuals, 7 trials, tilapia: 2 individuals 6 trials), but we chose to only include the data of one individual per species in the Figure 3 and in the illustration Figure 4, to ease the readability. We added a figure in the supplementary material of Figure 3, to illustrate the equivalent of Figure 3 for the other individual of both carp and tilapia (Figure 3 – supplement 1). We see that the results are consistent with what was reported in the previous version of the manuscript.

For the Skeletal kinematics analysis, one individual of tilapia had to be dismissed because most of the bone markers were consistently out of the field of view at the end of the food intake. This was specified in the text.

Please, note that all the data are available in the R code provided as a package in the supplementary material.

In the Methods, the rationale for using just 2 individuals, as compared to the more typical 3-4 as a minimum, is missing. In the Discussion, a few sentences should clarify how the low number of individuals may limit the interpretation and generalisability of the data. Given the total number of individuals across both species is 4, the overall conclusions supported across both species and both individuals are considered strongest.

A sentence was added to the Method section, explaining that two individuals per species are deemed sufficient, and therefore an appropriate choice when taking ethical considerations into account to reduce the number of experimental animals to a minimum (3R framework). Quantifying individual variability would require a higher number of individuals per species and is, therefore, not a goal of our study. The fact that both show the same result in the framework of the hypotheses from Figure 1 confirms this choice.

We added a sentence mentioning this limitation, yet adding that the consistency of the results tends to strengthen our conclusions:

“We reduced the number of experimental animals to a minimum for ethical considerations. As quantifying individual variability would require a higher number of individuals per species it was not a goal of our study, therefore intraspecific variation is not covered sufficiently by our analysis to allow generalizations within each species. Additionally, the limited number of species studied do not allow us to generalize our findings to other adult, omnivorous, teleost fish. Even if the number of analyzed individuals was limited to two per species, which may limit the interpretation of the data in the context of the STRANGE framework, the total number of individuals across both species (4 individuals), the number of trials (carp: 7 trials, tilapia: 6 trials), and the consistency of the results, imply that the mainly qualitative conclusions on overall patterns in water flow and skeletal motion are well supported.”

Finally, please keep in mind that the calculation of a biologically meaningful average and standard deviation representative for a species requires a minimum of three individuals.

We pooled the data from the two individuals of each species, but standard deviations are “among trials within species” and not “among individuals”. We made that clearer in the manuscript.

The major limitations remain (after processing all trials for all individuals across the two species to resolve the minor limitations) and impact the ability to conduct statistical analyses, inferences and conclusions that are biologically meaningful. Yet, this context is not sufficiently integrated in the presentation of the experimental design and discussion of the results.

We added a sentence in the Material and Methods to highlight the small number of individuals per species and the potential limitations it can cause in terms of generalizability of the results.

(5) The tracking of particles into the mouth should provide qualitative data that can highlight the hydrodynamics of the flows, but the speed of intra-oral flows is not fully examined. The manuscript is missing a quantitative estimate of the volume of water that passes near the esophagus, and of the kinematics of bone movements and how they correspond to fluid flows.

To quantify the volume of engulfed water that passes by the pharyngeal jaws, unfortunately, we would need more particles distributed over a larger volume in front of the mouth. Currently, the particles are distributed over a circular area surrounding the food at the bottom of the aquarium. This does not allow a quantification of the full water volume entering the mouth, and therefore also not the volume of water passing near the oesophagus. Future studies may further optimise our protocol to get a more complete view of what happens intra-orally with the entire water parcel entering the mouth. This will, however, require X-ray-visible particles that are in suspension for a long time, which would be technically very challenging to build.

Therefore, we would like to see quantitative mapping of the relationship between the distance from the mouth at the beginning of the strike to where the particle ends, and other quantitative measures. The authors also mention efficiency but make no attempt to quantify it. For the reviewers, it is the quantitative insight into these and other suction feeding parameters that has the potential to make this article particularly impactful.

We added a figure in the supplementary material (Figure 4 —figure supplement 2), showing the tracer trajectories, coloured according to their initial position. Generally, the particles tend to stay on the same side as their initial location, but there can be a change of side before the particles arrive close to the oesophagus. A Kendall test of correlation revealed a significant correlation in carp but not in tilapia (carp: p-value = 0.000287; tilapia: p-value = 0.07952).

As we did not find a solution to clearly define a metric of efficiency for suction feeding that can be quantified with our data, we no longer mention it.

(6) The beautifully illustrated tracks are a strength of the manuscript, but also mask some mechanistic details we would like to see dissected. The tracks in the current plot format confound space and time, meaning that the particles potentially started at different distances under different angles from the mouth and may have also started to move substantially at different times. We therefore believe additional figures are needed to quantitatively analyze this. Quantification will also enable the authors to make statements such as "food items that are initially somewhere close to the mouth…" as specific as possible based on the dataset.

Indeed, as the number of particles, their initial location relative to the individual, and the fish behaviour (angle, velocity, etc.) and anatomy (size, shape of the buccal cavity) differed between trials and individuals, we chose to use a representation of the data using space instead of time, as a standardization technique.

We added a Figure showing the particle distribution for each trial of each individual, based on the moment they start to move toward the buccal cavity (Figure 4 – supplement figure 1). We found no correlation between the moment of ingestion and the distribution of the tracers in the buccal cavity. However, visually, it seems that early-sucked particles tend to stay in the medial plane of the buccal cavity, participating to the central jet, whereas the late ones tend to end up on the lateral parts of the buccal cavity.

(7) We are curious if the suction effort of the fish may have varied across the trials (videos). Such variation has been shown to be primal in determining the speed of the external flows, hence we believe it is worthy of inclusion in the analyses.

Considering our effort to standardize the set-up (food type, position, number of tracers, etc.) our goal was to minimize the variation between individuals and trials. Therefore, it would not be relevant to quantify the variability in term of suction effort in the context of this study. We agree, however, that further studies using the same methods should try to quantify this.

Note that since we are aware that flow velocities are depending on suction effort, which depends on the type of food that is offered, we focus on the spatial patterns in the flow in the light of the hypothesis that were put forward in Figure 1, and not on the velocity magnitudes.

(8) The manuscript somewhat oversells the study's novelty; please try to better delineate the contribution. (i) Briefly clarify the exact contribution of the new method in the context of existing literature; what existed and has been done, which aspects are added / new etc. For example, the water markers seem to represent a modified design of previously reported particle and food item tracking approaches. Neither the XROMM technique nor particle tracking nor neutrally buoyant particle tracking are entirely new; for example, tracking marked prey items (with radio opaque markers) into the mouth and esophagus using XROMM has been done previously. Both particle tracking and particle image correlation are established for a wide range of applications. XROMM is also established for 3D imaging and tracking in the body. The new advances should be presented in the context of the relevant experimental literature.

We agree this was confusing in the original version of the manuscript, especially in the sentence from the introduction. We rewrote it to:

“Here, we develop a new technique based on biplanar high-speed X-ray videography to quantify the 3D pathlines of intraoral water and combine it with existing methods to track food and quantify 3D skeletal motions.”

This should clearly separate the new contribution from the existing methods and make it clear that we see the tracking of water as a separate method in addition to the tracking of food. The latter has indeed been done previously in many 2D x-ray studies, and in more recent 3D biplanar X-ray studies. The revised paragraph in the discussion now reverts back to the x-ray particle tracking protocols that have been used in industrial settings (reference Drake et al., 2011).

(ii) Discuss previous CFD studies more extensively, and how the limitations of those studies precluded the insights achieved by the present study. Considering these internal flows were modelled extensively by the authors (others), there is a model and theoretical framework that predicts how the internal flow behaves. This rich CFD and literature context is missing in the presentation of Figure 1, the Introduction and Discussion.

We added a paragraph in the Discussion, a new figure (Figure 6), and a supplementary video (Figure 6 – video 1) summarizing the results from previous CFD studies as requested. Note, however, that to our knowledge, 3D CFD models including an outflow of water through the gill slits do not exist currently. Consequently, to study patterns of intra-oral flow, the CFD literature is not rich. We included results from the currently most advanced, dynamically expanding 3D model of the head of a sunperch from Thompson et al., 2018 in Figure 6a. We clearly noted the limitations of this work (i.e., assumption of no outflow through gill slits). In Figure 6b, the non-expanding model from Provini and Van Wassenbergh 2018 is presented. Both CFD studies are discussed in the light of the current experimental results.

(9) Please discuss the tracking particle physical properties consistently and correctly: there is confusion about the particles being neutrally buoyant versus (page 16) negatively buoyant; this should be clarified throughout. The PIV literature provides a useful tracking error framework for relating the effect of particle size and density ratio to the ability of particles to track the flow. These established error analyses should be integrated into the Methods section in which the method is outlined. This should include estimated limitations due to particle size and buoyancy imperfections limiting the ability of the particles to trace out the flow pattern of interest. With this context the reader can fully comprehend the significant advance the current study achieves, within the acceptable limitations given the challenges involved in this tracking approach. The technical discussion paragraph could comment on how future tracing particles could/should be improved.

We maintain our statement concerning the negatively buoyant particles, in the Material and Methods section of the previous version of the manuscript. We measured the density of 19 particles (figure 7d), but the particles we used in the experiment were the ones that could remain at the bottom of the water tank, waiting for the fish to swallow them. By definition they were negatively buoyant.

We had previously evaluated the effect of size and density on tracking performance of our particles using CFD but have now worked this out in detail and added this CFD analysis to the manuscript. The results are presented in a new Figure (Figure 8), which is further illustrated by a video (Figure 8 – video 1). It serves to quantify the limitations due to particle size and buoyancy imperfection.

The CFD results reassure that the finite size (1.4 mm diameter) and density (up to about 1050 kg m-3) of the current sample of particles (Figure 7) does not hinder a realistic assessment of the suction flows by fish. A small deviation of the path in the direction of gravity can be expected (Figure 8b), but this should be smaller than 1 mm even for the heaviest particles of 1050 kg m-3. Lag during flow acceleration and overshoot during deceleration for the slightly negatively buoyant particles was relatively small (Figure 8c) and therefore acceptable to describe general patterns of flow during suction feeding.

As suggested, we added a discussion on how to improve future tracer particles in terms of achieving less variability in density. Miniaturising the manufacturing protocol by Drake et al., 2011 has this potential, in theory.